# OpenReview forum: "Decoupling Safety into Orthogonal Subspace: Cost-Efficient and Performance-Preserving Alignment for Large Language Models"
_ICLR.cc/2026/Conference — Submitted to ICLR 2026_

### Official Review · Reviewer_cYMV · 2025-10-24

**Soundness:** 2
**Presentation:** 3
**Contribution:** 2
**Rating:** 2
**Confidence:** 4

**Summary:**

This paper investigates LoRA-based refusal training as a cost-efficient approach for safety alignment of large language models (LLMs). The authors argue that LoRA fine-tuning constructs a “safety subspace” that is largely orthogonal to the model’s intrinsic weight space, thereby enhancing safety (i.e., reducing jailbreak success rate) while degrading general capabilities less than full fine-tuning. The paper provides both a theoretical intuition based on SVD analysis of LoRA update matrices and empirical evaluations across several instruction-tuned models, including Qwen2.5-7B-IT, LLaMA3.1-8B-IT, and Mistral-7B-IT.

**Strengths:**

* The figures and tables are well-organized, making it easy to follow
* Experiments cover multiple instruction-tuned backbones (Qwen2.5-7B-IT, LLaMA3.1-8B-IT, Mistral-7B-IT)

**Weaknesses:**

* **Limited conceptual novelty.** Most of the paper’s conclusions are already well established in prior literature. Earlier works such as [1] have systematically shown that LoRA updates will forget less and better keep general ability. [2] has shown that safety subspace is usally low rank. This paper mainly repackages these known observations without introducing a new mechanism or deeper insight.

* **The “safety gap between lora & finetuning” result is weak .** The only somewhat new finding, that the difference between LoRA and full fine-tuning is smaller in the safety domain than in other domains (code, finance) is not  convincing enough. The training datasets are limited, and it is unclear whether this effect is general or simply dataset-specific.

* **Missing robustness analysis.** The paper lack investigation about how well the learned “safety subspace” persists under further finetuning.

* **theoretical analysis is a bit trival**. The theoretical part provides a simple SVD-based explanation of “orthogonality” between LoRA updates and the base model weights.  the analysis is largely descriptive and does not yield new insight into how a safety subspace is actually formed, what governs its geometry, or how it differs from other task-specific subspaces.

[1] LoRA Learns Less and Forgets Less

[2] Assessing the Brittleness of Safety Alignment via Pruning and Low-Rank Modifications

**Questions:**

In section 5.3, the paper compares the right singular vectors $V_0$ of the original weight matrix $W_0$ and $V_{\Delta}$ of the LoRA update $\Delta V$, , both truncated to $r$ dimensions. However,  I think $W_0$  is typically higher rank, truncating  $V_0$  to be the same as $V_{\Delta}$ seems not that reasonable?

---

> ### Author Response · Authors · 2025-11-21
> **Response-1**
>
> **Q1: Limited conceptual novelty. Most of the paper’s conclusions are already well established in prior literature.**
>
> A1: Thank you very much for your review. You may have some misunderstandings about the differences between our study and some other work. We elaborated on the main differences in General Response.
>
> - In brief, our work and **"LoRA Learns Less and Forgets Less"** focus on different domains, tasks, and analytical perspectives. Their experiments focus solely on mathematics and code, so the conclusions may not generalize to safety alignment. Moreover, their study is largely empirical, examining how the LoRA rank and the choice of transformer modules influence its ability to approximate full fine-tuning. In contrast, our work primarily analyze the advantages of LoRA over full fine-tuning in safety alignment through the lens of subspace orthogonality, which is more in-depth and has not been explored in prior research.
>
> - Although our study and **"Assessing the Brittleness of Safety Alignment via Pruning and Low-Rank Modifications"** both lie within the broader area of LLM safety, they address fundamentally different research topic. Their work mainly analyzes already aligned LLMs, aiming to preserve safety when further general or malicious fine-tuning. In contrast, our study focuses on safety alignment for instruct models, aiming at improving safety while maintaining general capabilities.
>
>
> **Q2: The “safety gap between lora & finetuning” result is weak . The only somewhat new finding, that the difference between LoRA and full fine-tuning is smaller in the safety domain than in other domains (code, finance) is not convincing enough. The training datasets are limited, and it is unclear whether this effect is general or simply dataset-specific.**
>
> A2:
> Thank you for review. Perhaps you have some misunderstandings about our experiment.
> The experiments in Section 5.4 are specifically designed to verify the unique advantage of LoRA-based training for safety alignment, compared with applying LoRA in other domains. Our results show that "LoRA updates in other domains (e.g. finance and code) cause more significant other general performance drops and exhibit higher subspace similarity than in safety domain", indicating that **the orthogonality between the transformation subspace of W0 and ΔW is more pronounced in the safety domain than in other domains**.
>
> We supplement experiments in the Legal and Role-Play domain to better illustrate the generalization of our findings. The detailed experimental results are as follows:
>
> **Safety and Performance Comparision**
> |                    | SaladBench（Safety） | MT-Bench | MMLU  | MATH-500 | HumanEval | Avg (General Performance) |
> |------------------------------------|--------------------|----------|-------|----------|-----------|---------------------------|
> | LLAMA3.1-8B-IT                     | 24.62              | 7.43     | 54.48 | 45.8     | 68.63     | 60.8025                   |
> | + LoRA-safety       | 0.24               | 7.31     | 53.49 | 45.6     | 68.4      | 60.1475                   |
> | + LoRA-finance      | 25.84              | 7.31     | 51.3  | 45.6     | 66.25     | 59.0625                   |
> | + LoRA-legal          | 23.46              | 7.11     | 51.53 | 43.6     | 65.67     | 57.975                    |
> | + LoRA-code         | 24.41              | 7.24     | 49.83 | 46.6     | 62.71     | 57.885                    |
> | + LoRA-RolePlay         | 26.55              | 7.11     | 48.75 | 43.2     | 64.76     | 56.952                    |
>
> **Top five subspace similarity values**
> |  | Top 1       | Top 2       | Top 3       | Top 4       | Top 5       |
> |------------------------------------|------------|------------|------------|------------|------------|
> | Safety-LoRA   | 0.0860     | 0.0812     | 0.0812     | 0.0791     | 0.0791     |
> | Finance-LoRA  | 0.1148     | 0.1065     | 0.1040     | 0.1015     | 0.1008     |
> | legal-LoRA      | 0.1272     | 0.1170     | 0.1020     | 0.0911     | 0.0975     |
> | Code-LoRA     | 0.1310     | 0.1114     | 0.0955     | 0.0890     | 0.0882     |
> | RolePlay-LoRA     | 0.1026 |	0.1017 | 0.0963 | 0.0936 | 0.0918 |

---

> ### Author Response · Authors · 2025-11-21
> **Response-2**
>
> **Q3: Missing robustness analysis. The paper lack investigation about how well the learned “safety subspace” persists under further finetuning.**
>
> A3:
> Regarding the preservation of the safety subspace, we also explored the following aspects within the scope of this study：
>
> （1）In Section 3.2, we discuss the effect of LoRA-based refusal training under a lifelong learning setting. LoRA maintains stable overall performance while providing consistent safety improvements, whereas full-parameter training leads to more severe performance degradation.
>
> （2）To aid understanding, we additionally provide **SVD analysis under the lifelong learning setting**, showing that at each step, **LoRA-based refusal training constructs a safety subspace consistently orthogonal to the intrinsic transformation space of the model obtained at last step**.
>
> **llama3.1-8b-instruct: ΔW vs. W₀ subspace similarity (maximum value reported)**
> | W0                                              | ΔW                                               | Top-1 Similarity |
> |-------------------------------------------------|-------------------------------------------------|----------------|
> | llama3.1-8b-instruct                            | llama3.1-8b-instruct + Safety LoRA (Step0)     | 0.0734         |
> | llama3.1-8b-instruct + Safety LoRA (Step0), merged | llama3.1-8b-instruct + Safety LoRA (Step1)     | 0.0708         |
> | llama3.1-8b-instruct + Safety LoRA (Step1), merged | llama3.1-8b-instruct + Safety LoRA (Step2)     | 0.0645         |
>
>
> **Q4: theoretical analysis is a bit trival**
>
> A4:
> Thank you for your review. We do not introduce new theory; instead, we aim at leveraging subspace orthogonality theory to explain why LoRA-based fine-tuning outperforms full fine-tuning for safety alignment. A more detailed explanation is provided in our General Response.
>
> The question of "how the safety subspace is formed" may require analyzing the gradient update directions during training. We believe this is an interesting direction for future exploration.
>
> Regarding how the safety subspace differs from subspaces of other specific tasks, we provide some insights in Section 5.4. Our key finding is that "LoRA updates in the finance and code domains cause more significant general performance drops and exhibit higher subspace similarity", indicating that the **orthogonality of the transformation subspace is more pronounced in the safety domain than in other domains**.
>
>
> **Q5: Discussion of SVD truncated dimensions**
>
> A5:
> Thank you for your review. We have discussed this issue in **Appendix H**, where we also consider larger truncated dimensions:
>
> - Firstly, figure 11 shows the top 12 singular values of ΔW under different LoRA ranks. We observe that the number of non-negligible singular values of ΔW matches exactly the LoRA rank. In other words, among the components of VΔ, only the top-r components (r = LoRA rank) contribute meaningfully to ΔW, while the remaining components have negligible contribution.
> - Figure 10 examines cases where the SVD rank (truncated dimensions of VΔ and V₀) exceeds the LoRA rank. The similarity between VΔ and V₀ above 0.1 only appears in components that contribute weakly to ΔW.  In our experiments, we also increased the truncated dimensions to 100, 300, and 500, and the results were consistent.
>
> Therefore, for LoRA-based refusal training, the **major components of ΔW are clearly orthogonal to major components of W₀**, supporting the robustness of our findings.

---

> ### Author Response · Authors · 2025-11-28
> **A Kind Reminder for Reading the Response**
>
> Dear Reviewer:
>
> We have thoroughly addressed the issues you raised. For your convenience, we provide a brief summary below:
>
> - **Consider the differences between the two works you mentioned and our study:** In brief, our work and "LoRA Learns Less and Forgets Less" focus on different domains, tasks, and analytical perspectives. Although our study and "Assessing the Brittleness of Safety Alignment via Pruning and Low-Rank Modifications" both lie within the broader area of LLM safety, they address fundamentally different problems. The former focuses on safety alignment for instruct models, whereas the latter investigates continual general fine-tuning on already aligned models.
>
> - **Consider the generalization of cross-domain analyses in Section 5.4:** We supplement experiments in the Legal and Role-Play domain to better illustrate the generalization of our findings.
>
> - **Considering that the learned “safety subspace” persists under further fine-tuning:** We show that LoRA-based refusal training preserves the safety subspace under lifelong learning by maintaining stable performance and repeatedly constructing an orthogonal safety subspace at each step.
>
> - **Consider the contribution of our study:** Our study primarily focuses on the LLM Safety domain. Our experiments in Section 5.4 also revealed differences between LoRA-based refusal-SFT in the safety domain and LoRA fine-tuning in other domains. More details can be found in our response and general response.
>
> - **Consider SVD truncated dimensions:** Experiments at the bottom of Page 19 (Fig. 10–11) demonstrate that the meaningful top-r components of ΔW are orthogonal to major components (the number >> r) in W₀, validating the robustness of our conclusions.
>
> If you have any further questions, please feel free to point them out. We would be happy to discuss them with you.

---

> > ### Comment · Reviewer_cYMV · 2025-11-28
> >
> > I thank the authors for their detailed response. However, after reviewing the response and other reviewers' comments, I am inclined to maintain my original score.
> >
> > 1. Novelty Concerns: I still have concerns about the novelty of this paper. Although the authors claim they focus on safety and theory, the theoretical analysis itself is relatively trivial and general. It does not explain why safety would behave differently from other domains.
> >
> > 2. Robustness to Subsequent Fine-tuning: To clarify my earlier concern about robustness: prior work has shown that safety training is fragile and can be compromised even by fine-tuning on benign tasks [1]. What I intended to ask is whether LoRA and full fine-tuning would have the same robustness to fine-tuning attacks if you further fine-tune the model on other domains.
> >
> > 3. For point 2, my original concern was also whether LoRA would consistently outperform full fine-tuning if the authors used different safety training sets. But now I also have questions about the cross-domain comparison. If I understand correctly, the authors aim to show that LoRA has advantages over full fine-tuning for safety alignment. this would require comparing the performance gap between LoRA and full fine-tuning across different domains, rather than comparing LoRA performance across domains.  The current results only show that fine-tuning on code/finance affects general performance more than safety fine-tuning. But cannot show lora is special in safety domain.
> >
> >
> > [1] Fine-tuning Aligned Language Models Compromises Safety, Even When Users Do Not Intend To!

---

> ### Author Response · Authors · 2025-11-29
> **Response**
>
> **Q1: Robustness to Subsequent Fine-tuning**
>
> A1:
> Thank you for your review. We additionally perform further benign fine-tuning for two aligned models: LLAMA3.1-8B-IT + LoRA-based refusal-SFT and LLAMA3.1-8B-IT + full refusal-SFT. We randomly sampled 4,000 examples from OpenOrca for training. The experimental results are as follows:
> | Method | WildJailbreak | SG-Bench (Jailbreak) | SaladBench |
> |---|---:|---:|---:|
> | LLAMA3.1-8B-IT + full refusal-sft | 0.05 | 0.01 | 0.02 |
> | LLAMA3.1-8B-IT + full refusal-sft + benign FT | 29.5 | 11.9 | 12.92 |
> | LLAMA3.1-8B-IT + LoRA-based refusal-sft | 0.45 | 0.07 | 0.24 |
> | LLAMA3.1-8B-IT + LoRA-based refusal-sft + benign FT | 13.15 | 5.59 | 8.82 |
>
> It can be observed that, regardless of the alignment approach, further benign fine-tuning leads to a rebound in safety risks. However, this **rebound is significantly weaker for models aligned via the LoRA-based method, indicating that LoRA remains more robust than full-parameter fine-tuning against fine-tuning attacks**.
>
> **Q2: Require comparing the performance gap between LoRA and full fine-tuning across different domains**
>
> A2:
> Thank you for your feedback. We have supplemented the comparison between LoRA and full fine-tuning in the finance and code domains. The experimental results are as follows:
> | Method | MT-Bench | MMLU | MATH-500 | HumanEval |
> |---|---:|---:|---:|---:|
> | LLAMA3.1-8B-IT | 7.43 | 54.48 | 45.8 | 68.63 |
> | LLAMA3.1-8B-IT + Full-safety | 6.06 | 9.34 | 32.6 | 66.28 |
> | LLAMA3.1-8B-IT + Full-finance | 7.14 | 46.05 | 46.6 | 66.89 |
> | LLAMA3.1-8B-IT + Full-code | 6.93 | 47.65 | 46.8 | 61.37 |
> | LLAMA3.1-8B-IT + LoRA-safety | 7.31 | 53.49 | 45.6 | 68.4 |
> | LLAMA3.1-8B-IT + LoRA-finance | 7.31 | 51.3 | 45.6 | 66.25 |
> | LLAMA3.1-8B-IT + LoRA-code | 7.24 | 49.83 | 46.6 | 62.71 |
>
> We observe that
> - full-parameter domain-specific fine-tuning leads to a degradation of the model’s general capability in all cases, but this degradation is most severe in the safety domain.
> - In contrast, domain-specific LoRA fine-tuning also incurs some loss of general performance, but the loss in the safety domain is paradoxically the smallest.
>
> This highlights a unique advantage of LoRA in the safety domain — it is significantly more effective at mitigating the loss of general capabilities. As we have also demonstrated in our response to Reviewer u1tT, safety-related knowledge has substantially less overlap with the model’s intrinsic general capabilities compared to knowledge from other domains.
>
> **Q3: Novelty Concerns**
>
> A3:
> Regarding the differences between safety and other domains, we discuss this in Section 5.4 and our response of Q2. Our findings can be summarized as follows:
> - Full-parameter fine-tuning in the safety domain causes a significantly larger degradation in general performance than full fine-tuning in other domains.
> - In contrast, LoRA fine-tuning in the safety domain mitigates performance loss more effectively than LoRA  in other domains.
> - Subspace similarity analysis between W0 and ΔW shows that the orthogonality of the transformation subspace is more pronounced in the safety domain.
>
> Regarding theoretical contributions, we provide detailed clarification in the General Response. We do not propose a new theory; rather, we apply orthogonal subspace theory to the context of safety alignment and reveal why, in this special domain, LoRA achieves better safety alignment with minimal performance degradation compared to full fine-tuning.

---

### Official Review · Reviewer_iG1W · 2025-10-29

**Soundness:** 3
**Presentation:** 3
**Contribution:** 2
**Rating:** 4
**Confidence:** 3

**Summary:**

The paper proposes using LoRA-based finetuning as a "safety patch" for LLMs. The central claims are (1) efficacy: LoRA-based fine-tuning can achieve strong safety alignment while preserving general performance; (2) efficiency: lora is more cost-efficient than full-parameter fine-tuning; (3) mechanism: the paper argues this works because LoRA decouples the safety update into a low-rank subspace that is "largely orthogonal" to the original model's "intrinsic transformation".

**Strengths:**

- Cost-efficiency and simplicity: The method is presented as highly cost-efficient because it successfully achieves safety alignment by training solely on safety-critical data and also utilize lora technique.
- The paper demonstrates that LoRA-based SFT can substantially enhance model safety.

**Weaknesses:**

- The authors claim the theoretical insight as a part of the contributions. However, the concept of using orthogonal subspaces to prevent catastrophic forgetting in adapter-based tuning is not new. The paper itself cites O-LoRA, a method explicitly designed for "Orthogonal Subspace Learning" in continual learning. The core idea of this paper that a new task can be learned in a subspace orthogonal to the original model's to prevent interference is exactly the central thesis of O-LoRA. The paper fails to adequately differentiate its core mechanism from this prior work.
- One experiment is missing from the paper, that is LoRA with safety-general data. It is possible that LoRA-based SFT on the safety-general would perform even better, which would invalidate the paper's claim that mixing general data compromises safety.
- Regarding the life-long learning experiment, the authors seem to choose a rather weak baseline as DPO + full parameters seems to lead to low performance in a variety of tasks.
- There are also some related works that the authors could cite, for example [1,2]

[1] Refusal in Language Models Is Mediated by a Single Direction
[2] LoX: Low-RankExtrapolation Robustifies LLMSafetyAgainst Fine-tuning

**Questions:**

Please refer to the the weakness section.

---

> ### Author Response · Authors · 2025-11-21
> **Response**
>
> **Q1: The concept of using orthogonal subspaces to prevent catastrophic forgetting in adapter-based tuning is not new. The paper fails to adequately differentiate its core mechanism from this prior work.**
>
> A1:
> Thank you very much for your review. You may have some misunderstandings about the differences between our study and O-LoRA, as well as the main contributions of our works. We elaborated on these points in General Response.
>
>
> **Q2: One experiment is missing from the paper, that is LoRA with safety-general data.**
>
> A2:
> Thank you for your suggestion. We have added experiments on **LoRA with safety-general data**, and the results show that mixing general data significantly weakens the safety improvement and even introduces degradation in general performance.
>
> | Model Variant                                      | WildJailbreak ↓ | SG-Bench (Jailbreak) ↓ | SaladBench ↓ | MT-Bench ↑ | MMLU ↑ | MATH-500 ↑ | HumanEval ↑ |
> |----------------------------------------------------|------------------|-------------------------|--------------|------------|---------|-------------|--------------|
> | Qwen2.5-7B-IT + LoRA-Refusal-SFT (“safety-only”)   | 7.8              | 0.85                    | 1.24         | 7.64       | 69.47   | 70.2        | 81.16        |
> | Qwen2.5-7B-IT + LoRA-Refusal-SFT mixture (“safety-general”) | 9.05             | 2.54                    | 2.88         | 6.7        | 67.98   | 61          | 73.38        |
> | LLAMA3.1-8B-IT + LoRA-Refusal-SFT (“safety-only”)  | 0.45             | 0.07                    | 0.24         | 7.31       | 53.49   | 45.6        | 68.4         |
> | LLAMA3.1-8B-IT + LoRA-Refusal-SFT mixture (“safety-general”) | 13.55            | 5.19                    | 6.24         | 7.05       | 60.42   | 45          | 64.54        |
>
> Our SVD analysis further confirms this observation:
> after mixing general data, the transformation subspace constructed by LoRA-based fine-tuning no longer models safety exclusively, which increases its overlap with the model’s intrinsic transformation space.
>
> **Top five subspace similarity values**
> | Model & Setting                                     | Top 1 | Top 2 | Top 3 | Top 4 | Top 5 |
> |-----------------------------------------------------|-------|-------|-------|-------|-------|
> | **Qwen2.5-7B-IT + LoRA-Refusal-SFT ("safety-only")**      | 0.12  | 0.11  | 0.08  | 0.08  | 0.07  |
> | **Qwen2.5-7B-IT + LoRA-Refusal-SFT mixture ("safety-general")** | 0.23  | 0.21  | 0.13  | 0.12  | 0.10  |
> | **LLAMA3.1-8B-IT + LoRA-Refusal-SFT ("safety-only")**      | 0.09  | 0.08  | 0.08  | 0.08  | 0.08  |
> | **LLAMA3.1-8B-IT + LoRA-Refusal-SFT mixture ("safety-general")** | 0.14     | 0.13     | 0.12    | 0.12     | 0.11     |
>
> This provides additional evidence that **pure safety-focused LoRA training is crucial for maintaining orthogonality and preserving the model’s original capabilities**.
> Therefore, **LoRA with safety-only data remains the best current practice**, as it achieves strong safety improvements while preserving the model’s original capabilities.
>
>
> **Q3: Regarding the life-long learning experiment, the authors seem to choose a rather weak baseline as DPO + full parameters seems to lead to low performance in a variety of tasks.**
>
> A3:
> Thank you for your review. There may have been some misunderstandings. Actually, **full-parameter DPO (“safety-only”)** achieves the best overall balance between safety and performance among all full-parameter training methods in Section 3.1 (Table 1). Therefore, we use it as the baseline for the lifelong learning setting.
>
> We summarize the results in Table 1 as follows:
>
> - **Full-parameter DPO (“safety-general”)** preserves general capabilities well, but its safety improvement is relatively weak.
> - **Full-parameter SFT (“safety-only”)** provides the strongest safety improvement, but it causes severe degradation in certain general abilities (e.g., MATH, MMLU).
> - **Full-parameter SFT (“safety-general”)** performs similarly to full-parameter DPO (“safety-only”) in both safety and general capabilities, but does *not* outperform it, and it additionally requires mixing a non-trivial amount of general data at every step of life-long training.
>
> Therefore, we think that selecting full-parameter DPO (“safety-only”) as the baseline is reasonable, as it represents the strongest full-parameter competitor.
>
>
> **Q4: There are also some related works that the authors could cite**
>
> A4:
> Thank you for the suggestions, and we will include these references in the revised version. For a comparison between our work and the two work you mentioned, please refer to our General Response.

---

> > ### Comment · Reviewer_iG1W · 2025-11-26
> >
> > I thank the authors for the responses. I will update my score to a more positive one and I hope to see the requested changes / updates on related literatures in the final revision.

---

> > > ### Author Response · Authors · 2025-11-27
> > >
> > > Thank you for your response. We will update the manuscript based on the discussion.

---

### Official Review · Reviewer_u1tT · 2025-10-30

**Soundness:** 3
**Presentation:** 3
**Contribution:** 2
**Rating:** 2
**Confidence:** 3

**Summary:**

The paper conducts extensive experiments to investigate and show that Low Rank Adaptation (LoRA) only using a refusal dataset is a better and more reliable approach for weight-space safety alignment/tuning compared to full finetuning (SFT) with a mixed dataset of refusal and general-purpose data. Experiments are mostly focused on supporting this claim by showing that LoRA-based safety-tuning induces parameter updates that influences safety-related input more than general-purpose tasks (hence, less degradation of general utility).

**Strengths:**

1. **Writing:** Paper is well-organized with plenty details while maintaining coherent logical flow.
2. **Motivation:** The problem being addressed is well-motivated and Section 2 sets up the stage by quantifying how uncareful safety-tuning degrades general performance.
3. **Extensive empirical analysis:** Convincing empirical analysis is provided that LoRA is more reliable for the task by comparing and visualizing $\Delta W$ and $W_0$ using various matrix alignment metrics.

**Weaknesses:**

1. Although a plenty of prior works from safety alignment literature are brought up and discussed, some works that are directly related to LoRA-based safety alignment seem to be overlooked. Particularly, [1] also shows that LoRA on refusal dataset can bypass the "safety tax" (negative impact on general utility) in the case of reasoning abilities. Also, I think [2] takes it further by implementing an extra projection for LoRA updates to ensure it lies in the subspace present in the aligned model parameters.
2. The theoretical explanation also remains quite generic. To be more precise, the approximate orthogonality of $\Delta V$ and $V_0$ is a necessary condition for the observed experimental results. It would be more meaningful contribution if, for example, Figure 8(b) could be explained through this argument. *"Why does safety tuning induce more orthogonal parameter updates compared to finance and code?"*, which I would guess it is because the pre-training dataset had finance and code-related data already.

**Questions:**

Stemming from the weaknesses section:

1. Could authors compare and pinpoint portions of their contributions complementary/orthogonal to [1] and [2]? I admit that [1] is only recently publicly available (first arXiv version on July 22), while [2] is old enough and peer-reviewed.
2. Could authors comment on the pretrained models and validate/invalidate my guess in Weakness 2? Easiest way to test it would be to reproduce Figure 8(b) with a safety-aligned model as a baseline, and compare safety, finance, and code cosine similarities.

___

Overall, I find the investigations in this paper thorough and useful in most aspects. However, lack of comparison with directly related works makes it hard for me to place the paper in the map of current safety research state. I am open to change my score if both of my questions are addressed in a satisfactory manner.

___

### References

1. Yihao Xue, Baharan Mirzasoleiman. LoRA is All You Need for Safety Alignment of Reasoning LLMs. arXiv:2507.17075
2. Chia-Yi Hsu, Yu-Lin Tsai, Chih-Hsun Lin, Pin-Yu Chen, Chia-Mu Yu, Chun-Ying Huang. Safe LoRA: The Silver Lining of Reducing Safety Risks when Finetuning Large Language Models. NeurIPS 2024

---

> ### Author Response · Authors · 2025-11-21
> **Response**
>
> **Q1: Although a plenty of prior works from safety alignment literature are brought up and discussed, some works that are directly related to LoRA-based safety alignment seem to be overlooked.**
>
> A1:
> Thank you very much for your review. We have detailed the differences between the works you mentioned and our study in General Response.
>
> It is important to emphasize that
> - "LoRA is All You Need for Safety Alignment of Reasoning LLMs" is a contemporaneous work of ours, also submitted to ICLR 2026. Their study focuses on reasoning models and empirically investigates LoRA best practices in safety alignment (e.g., rank selection, layer choice). In contrast, our work targets instruction models and offers more in-depth empirical and theoretical analyses of why LoRA offers safety alignment advantages over full fine-tuning from subspace orthogonality and parameter variation perspective.
> - "Safe LoRA" focuses on **performing general fine-tuning on already-aligned LLMs** while preserving their safety. In contrast, our method addresses **how to perform safety alignment on instruction-tuned models**. These are fundamentally different topics.
>
> **Q2: The theoretical explanation also remains quite generic. It would be more meaningful contribution if, for example, Figure 8(b) could be explained through this argument.**
>
> A2: Thank you for your review. In fact, we indeed use subspace orthogonality theory to explain the experimental results in Section 5.4. Our finding was that **"safety subspace is uniquely orthogonal and less intrusive"**, a point also mentioned by Reviewer LmEX.
>
> Regarding the experiments in Section 5.4, I summarize the key points as follows:
> (1) Figure 8(b) shows the inner product between VΔ and V0 after LoRA-based fine-tuning across different domains (safety, finance, and code).
> (2) By comparing Figures 8(a) and 8(b), we observe that **the more orthogonal ΔW is to W0, the less the fine-tuning interferes with the model’s original capabilities**.
>
> We welcome any further discussion if our understanding of your concern is inaccurate.
>
> **Q3: "Why does safety tuning induce more orthogonal parameter updates compared to finance and code?", which I would guess it is because the pre-training dataset had finance and code-related data already.**
>
> A3:
> Thank you for your insightful suggestion. This is indeed a reasonable hypothesis.
> We further supplement the following experimental results: we applied LoRA fine-tuning to the LLAMA3.1-8B-IT model using data from different domains and observed the performance changes in each domain.
> |                           | SaladBench（Safety） | HumanEval（Code） | FinEval（Finance） |
> |-----------------------------|----------------------|-------------------|---------------------|
> | LLAMA3.1-8B-IT              | 24.62                | 68.63             | 31.89               |
> | LLAMA3.1-8B-IT + LoRA-safety | 0.24                 | 68.40             | 31.50               |
> | LLAMA3.1-8B-IT + LoRA-finance | 25.84                | 66.25             | 34.20               |
> | LLAMA3.1-8B-IT + LoRA-code   | 24.41                | 62.71             | 29.70               |
>
> - **Safety domain:** the performance improvement is most significant.
> - **Finance domain:** the improvement is relatively modest.
> - **Code domain:** no improvement.
>
>
> We infer that the initial model (LLAMA3.1-8B-IT) has already learned substantial knowledge in the code and finance domains, so additional fine-tuning in these areas contributes little to learning new information.

---

> ### Author Response · Authors · 2025-11-27
> **A Kind Reminder for Reading the Response**
>
> Dear Reviewer:
>
> We have thoroughly addressed the issues you raised. For your convenience, we provide a brief summary below:
>
> - **Consider the differences between the two works you mentioned and our study:** Our work targets instruct models and investigate why LoRA enables better safety alignment than full fine-tuning from subspace orthogonality and parameter variation perspectives, differing from SafeLoRA’s goal of preserving safety of safety-aligned models after further general fine-tuning, and from the contemporaneous “LoRA is All You Need” which takes a different research perspective and target models.
>
> - **Consider the explanation of Figure 8(b):** By comparing Figures 8(a) and 8(b), we observe that the more orthogonal the subspace of ΔW is to that of W0, the less the fine-tuning interferes with the model’s original capabilities, which further supports our conclusion regarding the orthogonal subspaces introduced by LoRA-based refusal SFT.
>
> - **Consider your insightful hypothesis:** We also validate through additional experiments that the initial model (LLAMA3.1-8B-IT) has already acquired substantial knowledge in the code and finance domains, such that further fine-tuning in these areas adds little new information. This also indicates that safety overlaps less with the model’s intrinsic capabilities than Code and Finance, and therefore LoRA-based Refusal-SFT has a smaller impact on the model’s original performance.
>
>
> If you have any further questions, please feel free to point them out. We would be happy to discuss them with you.

---

### Official Review · Reviewer_LmEX · 2025-11-06

**Soundness:** 4
**Presentation:** 4
**Contribution:** 4
**Rating:** 8
**Confidence:** 4

**Summary:**

This paper investigates the use of LoRA-based Refusal-training for safety alignment in Large Language Models (LLMs), proposing it as a cost-efficient, performance-preserving, and plug-and-play alternative to traditional full-parameter fine-tuning methods. A key finding is that LoRA-based alignment, even when trained solely on safety-critical data, significantly enhances safety (near-zero Attack Success Rate or ASR) while incurring minimal degradation to general capabilities (performance-preserving).
The authors provide a theoretical explanation based on transformation subspace orthogonality. They propose that the LoRA-induced safety update (ΔW) lies in a low-rank subspace that is largely orthogonal to the original model's intrinsic transformation space (W_0 ). This orthogonality, quantified using Sim(V_Δ,V_0)=V_Δ^⊤ V_0 ≈0 , minimizes the interference (or "catastrophic forgetting") between the safety adjustments and the model's inherent knowledge and abilities.

Empirical analyses support this claim through comparisons of:
1.Parameter Update Magnitude: Counter-intuitively, LoRA produces larger parameter updates than full-parameter training in most layers, yet it better preserves general performance.
2. Layer-wise Hidden State Shifts: LoRA-based Refusal-SFT induces smaller hidden state shifts on benign inputs but larger shifts on jailbreak attacks compared to full-parameter methods.
3. Orthogonality: LoRA-based alignment yields the lowest similarity (highest orthogonality) between the safety subspace and the model's intrinsic transformation space.
Furthermore, the paper demonstrates LoRA’s utility for lifelong safety alignment in multi-round red-teaming and shows that the safety subspace is more orthogonal and less intrusive than subspaces induced by domain-specific fine-tuning (e.g., code and finance).

**Strengths:**

*Originality & Significance: The paper presents a novel and highly effective method for safety alignment using LoRA-based Refusal-SFT and introduces subspace orthogonality as a powerful theoretical lens to explain the mechanism. This theoretical perspective is a significant conceptual contribution to understanding parameter-efficient fine-tuning (PEFT) in the context of safety and catastrophic forgetting.
*Quality & Clarity: The work is supported by rigorous and extensive empirical validation across multiple LLM architectures (Qwen, LLAMA, Mistral) and alignment paradigms (SFT, DPO). The results consistently and strongly support the core hypothesis, particularly the superiority of LoRA's safety-utility trade-off shown in Figure 1(a). The writing is excellent, making the complex technical concepts (SVD, orthogonality) accessible.
*Practical Value: The demonstration that LoRA-based Refusal-SFT performs best using only safety data is a crucial finding for practical deployment, as it eliminates the costly and difficult task of searching for the optimal proportion of safety-critical and general-purpose data. The plug-and-play feature for lifelong alignment is highly relevant for continuous model maintenance.

**Weaknesses:**

*Orthogonality Measurement Robustness: The current measure of orthogonality, Sim(V_Δ,V_0)=V_Δ^⊤ V_0 , relies on the ΔW from SVD, which itself is an approximation for LoRA weights (which are already low-rank AB matrices). While the results in Appendix H show a clear link between LoRA rank and the number of non-negligible singular values, a clearer theoretical or empirical connection between the inherent LoRA structure (ΔW=AB) and the guarantee of orthogonality to W_0 would strengthen the claim. The current finding is empirical; a theoretical bound or analysis on the orthogonality of span(V_Δ) for LoRA updates is missing.
*Generalizability of Safety Domain: The cross-domain analysis in Section 5.4 only compares safety with code and finance domains. To fully support the claim that the safety subspace is uniquely orthogonal and less intrusive, it would be beneficial to compare against other, potentially less "entangled" domains such as factual knowledge updates or style transfer.
*LoRA Rank Selection and Scaling: Although Appendix F and G discuss the effect of LoRA rank, the best rank for larger models (rank 16 for Qwen2.5-14B-IT) suggests the optimal rank is model- and size-dependent. A systematic study or guideline for selecting an appropriate LoRA rank, perhaps based on the initial W_0  properties (e.g., singular value decay of W_0 ), would improve the methodology's completeness and practical application.

**Questions:**

1. Theoretical Link to LoRA Structure: Can the authors provide a more formal theoretical justification, beyond empirical SVD, that the specific structure of LoRA updates (ΔW=AB) inherently biases the resulting subspace span(V_Δ) towards orthogonality with the initial weight matrix subspace span(V_0)? This would significantly elevate the theoretical contribution from an observation to a mechanism.

2. Orthogonality vs. Interference in Practice: The derivation in Appendix I (Non-Orthogonal Case) shows that the interference terms are
W_0x_Δ​+ΔWx_0 . The paper claims that LoRA-based SFT produces larger weight updates (∣ΔW∣) but smaller hidden state shifts (Δh^(l) ) on benign inputs. Since the hidden state shift is directly related to the interference, can the authors specifically quantify and compare the magnitude of the interference terms (∣∣W_0x_Δ+ΔWx_0∣∣) between LoRA-based and full-parameter models on benign inputs? This would directly connect the theoretical explanation to the key empirical findings.

3. Impact of Initial Model Alignment: The paper focuses on aligning instruction-tuned models (e.g., LLaMA3.1-8B-IT). How does the initial degree of safety alignment of the base model (e.g., a highly safe Llama-Guard-like model vs. a non-aligned base LLM) affect the final orthogonality of the LoRA safety patch? Does an already safe base model force the LoRA update to be more orthogonal, or does it make the effect negligible?

4. Beyond Refusal-SFT: The primary success is observed with LoRA-based Refusal-SFT. Given that DPO also exhibits better parameter and hidden state stability than SFT (Sections 5.1, 5.2) , why does LoRA-based DPO not achieve a similarly dominant trade-off as LoRA-based Refusal-SFT (e.g., lower safety gains for LoRA-based DPO in Table 1)? A deeper analysis into the mechanism difference between SFT and DPO loss functions in the orthogonal subspace framework would be insightful.

**Details Of Ethics Concerns:**

It utilizes jailbreak attacks (WildJailbreak, SG-Bench) and safety-critical data (SafeEdit-Train). While the work is intended to improve LLM safety, the use and potential public release of models/data trained on adversarial content, as well as the discussion of attack/defense methods, requires an ethics review to ensure adherence to responsible research practices and mitigation strategies are appropriately followed.

---

> ### Author Response · Authors · 2025-11-21
> **Response-1**
>
> **Q1: Orthogonality Measurement Robustness. The current finding is empirical; a theoretical bound or analysis on the orthogonality of span(V_Δ) for LoRA updates is missing.**
>
> A1:
> Thank you very much for your insightful review. We understand that you may feel performing SVD decomposition on ΔW with r = LoRA rank is only an approximation of ΔW=AB. It would be beneficial to examine the orthogonality between ΔW and W₀ in higher dimensions, which would make the orthogonality measure more robust.
>
> Actually, we have discussed this issue in **Appendix H**, where we also consider larger truncated dimensions of SVD to support the robustness of our findings:
>
> - **Figure 11** shows the top 12 singular values of ΔW under different LoRA ranks. We observe that the number of non-negligible singular values of ΔW **matches exactly the LoRA rank**. In other words, among the components of VΔ, only the **top-r components (r = LoRA rank)** contribute meaningfully to ΔW, while the remaining components have negligible contribution.
> - **Figure 10** examines cases where the SVD rank (truncated dimensions of VΔ and V₀) **exceeds the LoRA rank**. The similarity between VΔ and V₀ above 0.1 only appears in components that contribute weakly to ΔW. We also tried setting the SVD rank to 100, 200, and 300 (too many to include here due to space limitations), and the conclusions are consistent.
>
> We also believe that a more in-depth theoretical explanation can make the conclusions more solid, this will be our future work.
>
>
> **Q2: Generalizability of Safety Domain. It would be beneficial to compare against other, potentially less "entangled" domains such as factual knowledge updates or style transfer.**
>
> A2:
> Thank you very much for your suggestion. We are adding experiments in domains with potentially lower "entanglement" with model intrinsic performance, such as factual knowledge updating and style transfer. We selected 4000 <query, response> pairs in the RoleBench dataset, a role-playing fine-tuning dataset, for training LoRA for character style transfer.
>
> **(1)** The table below shows the impact of LoRA fine-tuning across different domains on the model's intrinsic general performance. As seen, safety LoRA causes the least disruption to intrinsic performance.
>
> **Safety and Performance Comparision**
> |                    | SaladBench（Safety） | MT-Bench | MMLU  | MATH-500 | HumanEval | Avg (General Performance) |
> |------------------------------------|--------------------|----------|-------|----------|-----------|---------------------------|
> | LLAMA3.1-8B-IT                     | 24.62              | 7.43     | 54.48 | 45.8     | 68.63     | 60.8025                   |
> | + LoRA-safety       | 0.24               | 7.31     | 53.49 | 45.6     | 68.4      | 60.1475                   |
> | + LoRA-finance      | 25.84              | 7.31     | 51.3  | 45.6     | 66.25     | 59.0625                   |
> | + LoRA-legal          | 23.46              | 7.11     | 51.53 | 43.6     | 65.67     | 57.975                    |
> | + LoRA-code         | 24.41              | 7.24     | 49.83 | 46.6     | 62.71     | 57.885                    |
> | + LoRA-RolePlay         | 26.55              | 7.11     | 48.75 | 43.2     | 64.76     | 56.952                    |
>
> **(2)** We also computed the subspace similarity between ΔW and W0, and observed that the safety LoRA-built safety subspace retains better orthogonality to the model's intrinsic transformation space.
>
> **Top five subspace similarity values**
> |  | Top 1       | Top 2       | Top 3       | Top 4       | Top 5       |
> |------------------------------------|------------|------------|------------|------------|------------|
> | Safety-LoRA   | 0.0860     | 0.0812     | 0.0812     | 0.0791     | 0.0791     |
> | Finance-LoRA  | 0.1148     | 0.1065     | 0.1040     | 0.1015     | 0.1008     |
> | legal-LoRA      | 0.1272     | 0.1170     | 0.1020     | 0.0911     | 0.0975     |
> | Code-LoRA     | 0.1310     | 0.1114     | 0.0955     | 0.0890     | 0.0882     |
> | RolePlay-LoRA     | 0.1026 |	0.1017 | 0.0963 | 0.0936 | 0.0918 |
>
>
> **Q3: LoRA Rank Selection and Scaling**
>
> A3:
> Thank you for your suggestion. In fact, as shown in **Appendix F (Figure 9)**, LoRA-based refusal training is not particularly sensitive to the LoRA rank hyperparameter.
>
> Under the current experimental conditions, we conclude that for the 7B model, rank=8 yields the best performance, while for the 14B model, rank=16 achieves the optimal performance. Our empirical results show that larger models require higher LoRA ranks to achieve optimal performance. Systematic larger-scale experiments will be explored in future work.

---

> ### Author Response · Authors · 2025-11-21
> **Response-2**
>
> **Q4: Theoretical Link to LoRA Structure**
>
> A4:
> We thank the reviewer for the suggestion. Intuitively, the LoRA update(ΔW = AB) is a low-rank perturbation constrained within the subspace spanned by the columns of \(A\) and rows of \(B\). Since \(A\) and \(B\) are initialized independently of the original weight matrix \(W_0\) and updated only through gradients on the safety-critical data, the resulting \(ΔW\) is naturally confined to directions largely uncorrelated with the dominant subspace of \(W_0\).
>
> Formally, **if the dominant singular vectors of \(W_0\) span a high-dimensional space and ΔW has rank r << rank(W_0)\, the probability that Span(VΔ) significantly overlaps with Span(V0) is low, leading to approximate orthogonality**. This provides a theoretical rationale for the empirical observations from SVD.
>
>
> A more rigorous analysis considering the gradient update directions during LoRA-based fine-tuning would provide deeper insight. We consider this a promising direction for future work.
>
>
> **Q5: Orthogonality vs. Interference in Practice: can the authors specifically quantify and compare the magnitude of the interference term**
>
> A5:
> Thank you very much for your careful review. We analyze 10 randomly selected samples from MT-bench. For each parameter matrix, we follow these steps:
>
> 1. Perform SVD decomposition on W0 and ΔW, and compute the components x0 and xΔ of the input x along the directions of V0 and VΔ.
> 2. Calculate W0 * xΔ + ΔW * x0 and compute its L2 norm.
> 3. For each weight matrix in the LLM, compute the norm of the interference term and record both the mean and maximum values.
>
> | Model | Interference  (mean) | Interference (max) |
> |-------|-----------------------------|----------------------------|
> | llama3.1-8b-instruct + LoRA-based Refusal-SFT | 2.863 | 13.66 |
> | llama3.1-8b-instruct + Full SFT ("Safety-only") | 5.002 | 31.41 |
> | llama3.1-8b-instruct + Full DPO ("Safety-only") | 5.021 | 34.72 |
>
> This confirms that LoRA shapes safety subspace has a smaller impact on the model's intrinsic transformation space, consistent with the findings in our paper.
>
>
> **Q6: Impact of Initial Model Alignment:  How does the initial degree of safety alignment of the base model affect the final orthogonality of the LoRA safety patch?**
>
> A6:
> Thank you very much for your careful review.
> We have discussed the impact of the initial model’s safety in Section 3.2. In the lifelong safety alignment setting, each step trains a new safety LoRA patch on top of the previously aligned model (after merging the prior LoRA weights). Table 2 presents a macro-level analysis of safety and model performance.
>
> We further conducted SVD and subspace orthogonality analysis by treating the parameters of the previously aligned model as W0 and the current step’s safety LoRA weights as ΔW. The results show that, **regardless of the initial model’s safety, the safety LoRA consistently yields a subspace approximately orthogonal to the model’s intrinsic transformation space**.
>
> ##### llama3.1-8b-instruct: ΔW vs. W₀ subspace similarity (maximum value reported)**
> | W0                                              | ΔW                                               | Top-1 Similarity |
> |-------------------------------------------------|-------------------------------------------------|----------------|
> | llama3.1-8b-instruct                            | llama3.1-8b-instruct + Safety LoRA (Step0)     | 0.0734         |
> | llama3.1-8b-instruct + Safety LoRA (Step0), merged | llama3.1-8b-instruct + Safety LoRA (Step1)     | 0.0708         |
> | llama3.1-8b-instruct + Safety LoRA (Step1), merged | llama3.1-8b-instruct + Safety LoRA (Step2)     | 0.0645         |
>
>
>
> **Q7: Beyond Refusal-SFT: Given that DPO also exhibits better parameter and hidden state stability than SFT. A deeper analysis into the mechanism difference between SFT and DPO loss functions in the orthogonal subspace framework would be insightful.**
>
> A7:
> We thank the reviewer for the careful feedback. While LoRA-based DPO preserves model performance, its improvement in safety is limited. We attribute this to the fact that **"DPO results in substantially smaller parameter updates than Refusal-SFT"**, as summarized in Section 5.1.
>
> For different safety alignment methods, both the magnitude and direction of Δ𝑊 influence the post-alignment model’s safety and general performance. For LoRA-based DPO, we conducted SVD on Δ𝑊 and computed its subspace similarity with 𝑊0 (top-1 similarity = 0.065 for llama3.1-8b-instruct). The results show that DPO still preserves orthogonality of the safety subspace, which explains why model performance is maintained. However, the smaller magnitude of
> Δ𝑊 limits the safety improvement.
>
> From the objective function perspective, the smaller parameter updates in DPO arise from its hyperparameter controlling the deviation between the new policy model and the reference model. We plan to further investigate this in future work.

---

### Official Review · Reviewer_Rd8i · 2025-11-10

**Soundness:** 3
**Presentation:** 3
**Contribution:** 3
**Rating:** 6
**Confidence:** 3

**Summary:**

The paper discusses how to implement safety finetuning for LLMs using LoRA weight updates. The authors argue that full-parameter safety finetuning on safety datasets alone still causes unnecessary changes in weight directions that are indicative of performances; fine-tuning on both safety and general data compromises safety guarantees. Thus, they hypothesize that performance and safety can be decomposed into (almost) orthogonal directions and using LoRA allows for safety finetuning to focus on the 'safety' directions only.

**Strengths:**

The paper addresses an important and timely topic. Already in this conference, there is another submission that also attempts to use LoRA for safety finetuning.

The experiments performed in the paper is comprehensive and well-demonstrates the points being made, which is that LoRA directions are generally almost orthogonal to base weight directions.

The proposed methodology is simple enough to be tested in practice with ultra-large LLMs, making the research valuable not just to academia but also to the industry.

**Weaknesses:**

The theoretical justifications of the paper is rather lacking in several ways:
- No orthogonal decomposition guarantees. The authors assumed that performance and safety can be orthogonalized (at least approximately) without giving arguments, even heuristical arguments, on why that should be the case. While experimental data shows that LoRA update has small (matrix) inner product with the base weights, that fact can be induced by implicit or explicit regularizations (since LoRA is constraint to have small rank, it must packs all gradient update information in a few dimensions; and this is done most efficiently with dimensions that are orthogonal to the base weights). In the above scenario, the orthogonality behavior may be at odd with 'true' safety dimensions (which was shown to exist, for instances, in Wei et al., "Assessing the brittleness of safety alignment via pruning and low-rank modifications"), and by forcing the update to be orthogonal via LoRA, safety guarantees are weakened.
- Orthogonality of weights does not imply independence of performance. While it is a point usually made in the literature that orthogonal updates to the weights preserve original performance, it is still a heuristical argument, since the final model is nonlinear in the transformer weights. It is fine to use this idea as a motivation and test empirically again (which the authors did a decent job empirically), the final results are inherently experimental in nature and the theoretical contribution would be too weak to be counted as a major contribution of the paper. This is also relevant to the next point, since the authors distinguish themselves from literature mostly by this 'orthogonalization' concept.
- Missing direct comparison to Wei et al  "Assessing the brittleness of safety alignment via pruning and low-rank modifications", which identifies 'safety directions' and 'safety neurons' quantitatively. Why, or under which condition, is the methodology in the current paper more correct than that proposed in Wei et al (pruning least safety-relevant neurons or removing safe-relevant directions (via LoRA) to improve safety)? While the reviewer think that there are enough differences between the two papers, a direct comparison should be done to previous LoRA methods for LLM safety.
- Minor: the authors have pointed out that different finetuning methods (DPO vs SFT-safety, etc.) induce different norms/magnitude in weight changes, yet the (dis)similarity score (matrix inner product of base and update weights) are left unnormalized. This may result in the small values observed simply coming from the matrix weights values being small, rather than orthogonality. Some normalization should be done in this comparison.

The weaknesses pointed out above are mainly conceptual and theoretical. Since the paper's impact and focus is purely experimental, I still recommend weak acceptance but would consider increasing my score if my concerns are addressed.

**Questions:**

See weaknesses

---

> ### Author Response · Authors · 2025-11-21
> **Response**
>
> **Q1: No orthogonal decomposition guarantees. The authors assumed that performance and safety can be orthogonalized (at least approximately) without giving arguments, even heuristical arguments, on why that should be the case.**
>
> A1:
> Thank you very much for your review. We believe there may be some misunderstanding about our work, and we hope the following clarification helps:
>
> Firstly, we do *not* assume safety and performance are inherently orthogonal; rather, our experiments aim to reveal this as a natural advantage of LoRA over full fine-tuning on safety alignment: "the principal components of LoRA-induced update ΔW is nearly orthogonal to those of the base model parameters W₀, enabling improved safety without harming performance". For more information on the differences between Wei et al.'s work and our study, please refer to our General Response.
>
> In addition, two details need to be clarified:
>
> **(1) Why ΔW represents a “safety direction”**
>
> Because LoRA-based refusal training uses *only* safety data (<malicious instruction, safe response> pairs), the update ΔW reflects parameter changes driven specifically by safety alignment, thus forming a meaningful safety subspace.
>
> **(2) On orthogonality via explicit or implicit regularization**
>
> - This study aims to reveal the advantages of LoRA-based refusal-SFT over full-fine-tuning in terms of safety alignment, and to provide sufficient experimental and theoretical explanations, rather than to propose a new regularization method.
> - Constraining gradient updates to remain orthogonal to base-weight directions is indeed an interesting idea, and we plan to explore it in the future.
>
> **Q2: Orthogonality of weights does not imply independence of performance. The final results are inherently experimental in nature and the theoretical contribution would be too weak to be counted as a major contribution of the paper.**
>
> A2:
> Thank you very much for your review. Firstly, we clarify the theoretical contributions in our study in General Response.
>
> Furthermore, we took into account the issue that the LLM itself is not a linear transformation during our experiments. To address this concern, we analyze **each weight matrix separately** (Wq, Wk, Wv, MLP proj_gate, proj_up, proj_down, etc.) throughout our experiments. This comprehensive, matrix-by-matrix analysis helps reduce experimental bias and ensures a more reliable evaluation.
>
> **Q3: Why, or under which condition, is the methodology in the current paper more correct than that proposed in Wei et al (pruning least safety-relevant neurons or removing safe-relevant directions (via LoRA) to improve safety)?**
>
> A3:
> First, regarding the differences between this study and the work of Wei et al., please refer to our General Response.
>
> In addition, I would like to clarify some details:
> Wei et al. do discuss two strategies for enhancing safety:  a) removing neurons or ranks deemed to have minimal impact on safety, and
> b) freezing safety-critical regions during general fine-tuning.
> However, The experiments in their paper show that the safety improvement is relatively weak, and the performance is unstable under different attack types, making it difficult to apply to large-scale model training.
>
> We argue that these approaches are more suitable for **lightweight pruning on already aligned models**.
> In contrast, our method is suitable for **safety alignment training for instruction tuned models**.
>
> **Q4: Some normalization should be done in this comparison.**
>
> A4:
> We use the cosine function from the official SciPy library.
> Since cosine similarity internally normalizes the input vectors, it is invariant to differences in vector norms, and therefore not affected by variations in weight magnitudes introduced by different fine-tuning methods.
> Below is our code:
> ```python
> from scipy.spatial.distance import cosine
>
> def cosine_sim(vec1, vec2):
>     v1 = vec1.flatten().numpy()
>     v2 = vec2.flatten().numpy()
>     return 1 - cosine(v1, v2)
> ```

---

> > ### Comment · Reviewer_Rd8i · 2025-11-25
> >
> > The reviewer thanks the authors for the rebuttal, in which the authors have addressed most relevant concerns and relevant discussions over related works have been added. However, a key issue has not been addressed, which is that the small absolute values (e.g. as reported in Figure 7b) of matrix inner product (for e.g. the last equation on page 6) does not mathematically imply that the subspaces are orthogonal, unless the matrices in question are normalized. While the authors, in the rebuttal, pointed out the use of cosine similarity (which is normalized), key equations for orthogonality, such as the last equation on page 6, is not normalized. To give an example of why this is a problem, 2 random matrices that are far from orthogonal subspaces can have extremely small inner product, if the entries themselves have small values.
> >
> > Since the manuscript was only slightly improved to add relevant details that should have been included in the full submission, I am maintaining my score.

---

> > > ### Author Response · Authors · 2025-11-25
> > > **Response**
> > >
> > > We sincerely thank the reviewer for pointing out this issue. This was an oversight in our writing. As mentioned in the rebuttal, the actual metric we use to evaluate the subspace relationship in our experiments is the cosine similarity, and the rigorous formulation should be as follows:
> > >
> > > $$
> > > \left(\frac{V_\Delta}{\|V_\Delta\|}\right)^T
> > > \left(\frac{V_0}{\|V_0\|}\right) \approx 0
> > > $$
> > >
> > > We have updated the manuscript accordingly (Page 7 and Page 20).

---

> > > > ### Comment · Reviewer_Rd8i · 2025-11-25
> > > >
> > > > I thank the author for quickly addressing the concern. The code indeed looks like it implements a normalized version.. However, in the write-up, please also fix your definition of sim(V_Delta, V_0) at the end of page 8. Also, in writing a norm as normalization, please specify the matrix norm (since there are a couple of reasonable choices) that you are using, or add a line explaining the normalization. It would be best to also argue/ablate on different norms to see if the results are the same but given the timeline, this is not a major point.

---

> > > > > ### Author Response · Authors · 2025-11-26
> > > > > **Response**
> > > > >
> > > > > Thanks for the careful review. We have fixed the definition of $\mathbf{Sim}(V_\Delta, V_0)\$ at the end of page 8 to align with the formula in Section 4. Additionally, we have included the statement: "In the experiment, we used L2 normalization." in the revised version.
> > > > >
> > > > > Since we focus on the directional differences between the subspaces corresponding to W0 and ΔW, cosine similarity is an appropriate metric. L2 norm is the standard approach for calculating cosine similarity.

---

### Author Response · Authors · 2025-11-21
**General Response**

We thank all reviewers for their thoughtful and constructive feedback. We are encouraged that reviewers found our work well-motivated (u1tT), methodologically novel and practical (LmEX, Rd8i), supported by solid and comprehensive experiments (LmEX, Rd8i, u1tT), theoretically valuable for safety alignment (LmEX), and clearly written (LmEX, u1tT).

The reviewers’ primary concerns center on two areas: (1) **comparison with related or concurrent works**, and (2) **the scope and clarity of our theoretical contribution**. We address both issues directly and in depth in the General Response below.

---

> ### Author Response · Authors · 2025-11-21
> **General Response**
>
> ## 1. Comparison of Some Related Works
> We sincerely appreciate the reviewer for highlighting these relevant and concurrent works, and we will add these citations in the revised version.
> In fact, our study differs significantly from these works, as summarized below:
>
> | Title | Domain | Target Model | Task | Core Contribution |
> |-------|------------|------------|--------|-----------------------------------------|
> | [1] O-LoRA | General | Instruct model | General fine-tuning | A method to mitigate catastrophic forgetting in multi-task continual learning. |
> | [2] LoRA Learns Less and Forgets Less | General | Instruct model | General fine-tuning | Empirical comparison between LoRA and full-parameter fine-tuning. |
> | [3] Assessing the Brittleness of Safety Alignment... | Safety | Safety-aligned instruct model | Fine-tuning attack & jailbreak attack | Empirical and theoretical analyses explaining why safety-aligned models are vulnerable to fine-tuning and jailbreak attacks. |
> | [4] Safe LoRA | Safety | Safety-aligned instruct model | Fine-tuning attack | A method enabling continuous general fine-tuning on already-aligned LLMs while preserving their safety. |
> | [5] LoX | Safety | Safety-aligned instruct model | Fine-tuning attack | Empirical and theoretical analyses explaining why safety-aligned models are vulnerable to fine-tuning attacks. |
> | [6] Refusal in LLMs Is Mediated by a Single Direction | Safety | Safety-aligned instruct model | Jailbreak attack | Empirical analysis reveals the inherent vulnerability of the safety-aligned models, and a novel white-box attack method is designed. |
> | [7] LoRA is All You Need for Safety Alignment of Reasoning LLMs | Safety | Reasoning model | Safety alignment | A method to perform safety alignment training on reasoning models while preserving reasoning capabilities. |
> | **Our Work** | Safety | Instruct model | Safety alignment | Empirical and theoretical analyses of the unique advantages of LoRA-based safety alignment over full fine-tuning on safety-performance trade-off. |
>
> **(1)** [1] and [2] study general-domain tasks, while we focus on safety alignment. Our aim is not to modify LoRA as [1], but to explain the unique advantages of LoRA for safety alignment compared to full training from the perspective of subspace orthogonality and parameter changes.
> **(2)** Although [3,4,5] also involve LoRA, their focus is on **fine-tuning attacks**—specifically, how to ensure that safety-aligned LLMs remain safe after further non-malicious or malicious fine-tuning. In contrast, our work targets the problem of **safety alignment** itself: how to train an instruction-tuned model that improves safety without degrading general capabilities. They are totally different.
> **(3)** [6] mainly proposes an attack method, while our study falls under the category of safety alignment/defense.
> **(4)** In addition, [7] is a concurrent work and is also an ICLR 2026 submission. Their study focuses on reasoning models and empirically explores best practices for LoRA-based fine-tuning, such as selecting LoRA rank and choosing the most important layers or modules to modify. Our work, by contrast, concentrates on instruct models and provides deeper analysis explaining why LoRA, compared with full fine-tuning, confers advantages for safety alignment.
>
> In summary, our work demonstrates that LoRA-based refusal training offers clear advantages over full fine-tuning for safety alignment: it achieves strong safety improvements using only safety data (without any general instruction data), while preserving model performance well. We further provide a detailed theoretical and empirical account of this effect through the lens of subspace orthogonality.
>
> ## 2. Clarification of the theoretical contributions in our study
> Our work does not introduce new theoretical formulations; instead, it **applies subspace orthogonality theory to the context of LLM safety alignment** to analyze how model parameters change before and after alignment.
>
> Through the derivations in Section 4 and Appendix I, we aim to construct an idealized formulation of the safety alignment objective.
> This serves two purposes:
> (1) It helps explain our empirical finding that **LoRA-based refusal training improves safety while preserving performance better than full-parameter fine-tuning**, and our experiments in Sections 5 are designed based on these theoretical insights.
> (2) It provides a theoretical perspective that may guide the development of future safety alignment algorithms.
>
> We also analyzed how LoRA-based tuning in safety differs from other domains (finance/code) (Section 5.4). We found that LoRA updates in those domains lead to larger general performance drops and higher subspace similarity, whereas safety tuning exhibits more pronounced subspace orthogonality. Therefore, we believe that the theoretical insight of LoRA in safety domain has unique value.

---

### Comment · Area_Chair_oNUw · 2025-11-23
**Next Steps Following Authors’ Rebuttal: Review Rebuttal and Participate in Discussion**

Dear Reviewers,

Thank you very much for your thoughtful evaluations of this paper.

Now that the authors have submitted their rebuttal, I kindly ask you to take the following steps (if you have not done so already):

- Read the other reviews as well as the authors’ response.
- Consider whether the rebuttal and additional comments affect your assessment of the paper.
- Engage in interactive discussion with the authors **before November 25**, encouraging a dynamic exchange rather than a one-sided rebuttal.

The current reviews for this paper are mixed. Your contributions at this stage are essential for forming a well-informed final decision. I therefore ask that you reassess your views in light of the authors’ responses and the broader discussion among reviewers.

I am happy to join and support the discussions between you and the authors. Please feel free to share your thoughts and participate actively in the discussion.

Thank you once again for your service to ICLR 2026.

Best regards,

 AC

---

### Meta-Review · Area_Chair_uvck · 2025-12-17

**Summary:**

Reviewers have discussed thoroughly the claimed novelty and depth, particularly whether framing LoRA-based safety alignment as learning an “orthogonal safety subspace” constitutes a substantive advance beyond prior work on low-rank adaptation, catastrophic forgetting, and safety directions.

Reviewers broadly agreed that the empirical study is thorough and carefully executed, and that the practical finding, e.g., LoRA enables safety alignment with less performance degradation than full fine-tuning, is useful and largely correct. However, many reviewers questioned whether the work moves beyond a refined empirical restatement of known PEFT behavior, especially given the largely descriptive and post-hoc nature of the subspace analysis. AC agrees on their sentiments.

**Reviewer Concerns:**

The rebuttal successfully addressed several concrete issues, including adding missing ablations, clarifying normalization, extending cross-domain comparisons, etc. However, the central concerns were only partially alleviated: reviewers remained unconvinced that the “orthogonal subspace” perspective offers a genuinely new mechanism rather than a repackaging of established intuitions from prior LoRA, O-LoRA, and safety-direction literature. In particular, the theoretical component was still viewed as largely heuristic, providing limited explanation for why safety should systematically behave differently from other domains, and falling short of yielding predictive or design-level guidance.

**Reviewer Scores:**

Based on the discussion, I would expect reviewer scores to remain mixed and largely near their original positions.

---

### Decision · Program_Chairs · 2026-01-26

Reject